# Cell Death in the Tumor Microenvironment: Implications for Cancer Immunotherapy

**DOI:** 10.3390/cells9102207

**Published:** 2020-09-29

**Authors:** Varsha Gadiyar, Kevin C. Lahey, David Calianese, Connor Devoe, Dhriti Mehta, Kristy Bono, Samuel Desind, Viralkumar Davra, Raymond B. Birge

**Affiliations:** Department of Microbiology, Biochemistry and Molecular Genetics, Cancer Center, Rutgers New Jersey Medical School, 205 South Orange Ave, Newark, NJ 07103, USA; vg310@gsbs.rutgers.edu (V.G.); kevcl@gsbs.rutgers.edu (K.C.L.); dcc139@gsbs.rutgers.edu (D.C.); cdevoe@bu.edu (C.D.); dhriti228@gmail.com (D.M.); kbb82@njms.rutgers.edu (K.B.); szd4@gsbs.rutgers.edu (S.D.); davravr@gsbs.rutgers.edu (V.D.)

**Keywords:** apoptosis, efferocytosis, compensatory proliferation, immune evasion, cancer, phosphatidylserine (PS)

## Abstract

The physiological fate of cells that die by apoptosis is their prompt and efficient removal by efferocytosis. During these processes, apoptotic cells release intracellular constituents that include purine nucleotides, lysophosphatidylcholine (LPC), and Sphingosine-1-phosphate (S1P) that induce migration and chemo-attraction of phagocytes as well as mitogens and extracellular membrane-bound vesicles that contribute to apoptosis-induced compensatory proliferation and alteration of the extracellular matrix and the vascular network. Additionally, during efferocytosis, phagocytic cells produce a number of anti-inflammatory and resolving factors, and, together with apoptotic cells, efferocytic events have a homeostatic function that regulates tissue repair. These homeostatic functions are dysregulated in cancers, where, aforementioned events, if not properly controlled, can lead to cancer progression and immune escape. Here, we summarize evidence that apoptosis and efferocytosis are exploited in cancer, as well as discuss current translation and clinical efforts to harness signals from dying cells into therapeutic strategies.

## 1. Introduction

### 1.1. Apoptosis and Efferocytosis Provide Homeostatic and Tolerogenic Signals Under Physiological Conditions

Under normal physiological conditions, it has been estimated that ~50–100 billion cells die per day, translating into 0.4–0.5% of the estimated 35 trillion cells in an average adult [1]. Under such physiological conditions, apoptosis is a homeostatic non-pathophysiological event, facilitating the removal of unwanted, aged, and genetically damaged cells, while allowing repopulation of new cells from precursor and/or stem cells [2,3]. In vivo, apoptotic cells are efficiently tagged for removal and the final physiological fate of apoptotic cells in vivo is their rapid removal by phagocytic cells (efferocytosis) [4,5]. Efferocytosis is a term from the Latin translation “efferre”, meaning to carry to the grave, that describes the biochemical and cell biological events by which apoptotic cells are recognized and processed by phagocytic cells, including either professional cells (macrophages or dendritic cells) as well as non-professional cells, such as epithelial cells, endothelial cells, and fibroblasts [6,7]. Efficient efferocytosis not only ensures a proper and dignified burial for the dying cells but, importantly, it spares the tissue microenvironment of decay and leakage of intracellular components of the dying cell, an event that can trigger acute and chronic inflammation and ultimately lead to autoimmunity and tissue damage [8,9]. Evidence for a critical role for efficient efferocytosis in autoimmunity is supported by both mouse knockout studies and in human autoimmune patients that show defective clearance and/or disruption of receptors that sense apoptotic cells which can lead to pathological states of chronic inflammation [10,11,12].

Immunologically, apoptosis and efferocytosis are termed “silent processes”, unfortunately, an awkward misnomer since it implies the absence of extant signaling and cell communication. By contrast, both apoptotic cells and efferocytic cells produce a complex myriad of autocrine and paracrine factors that are tightly regulated and evolutionarily conserved and interact extensively with their environment to coordinate a host physiological response [13]. These signals include factors that promote compensatory proliferation [14,15], tissue remodeling/maintenance of the extracellular matrix that resembles wound healing and repair [16,17], and immune skewing signals that suppress inflammation [2,18] and facilitate the resolution of inflammation [19,20] (Figure 1). While such events are important to preserve tissue function and prevent chronic inflammation, signals from both apoptotic cells and efferocytic cells are pathologically exploited in the cancer microenvironment to promote tumor progression and immune escape [21]. Further, enhanced apoptosis, clearly a major goal of targeted therapeutics and radiation oncology, if not properly managed, can have unanticipated consequences to enhance tumor progression and repopulation [22,23]. Here, we discuss evidence that apoptosis and efferocytosis can be pathological events that promote tumor progression and immune escape as well as recent clinical efforts to alter signals from dying and efferocytic cells with a focus on immuno-oncology.

### 1.2. Physiological Signals from Apoptotic and Efferocytic Cells Promote Compensatory Proliferation and Modify the Local Immune Microenvironment

Hallmarks of caspase-mediated apoptosis are characterized and defined by cell-intrinsic properties such as DNA laddering, condensation of chromatin, cytosolic shrinkage, and loss of mitochondrial membrane permeabilization. However, apoptotic cells also directly secrete chemoattractant factors and endogenous proliferative factors in a paracrine manner that alerts the neighboring local tissue microenvironment for clearance and regeneration. Purine nucleotides, such as ATP and UTP [24], as well as lysophosphatidylcholine (LPC) [25,26], act as short-range chemo-attractants that recruit immune cells (mostly myeloid-derived phagocytes such as macrophages and DCs). In the case for ATP and UTP, this process requires the caspase-mediated activation of pannexin 1 (PANX1) channels to release nucleotides from apoptotic cells and the subsequent nucleotide detection by purinergic receptors such as P2Y2 [27,28]. In addition, caspase-3-mediated activation of a calcium-independent phospholipaseA2 (iPLA2) leads to COX-2 dependent PGE2 upregulation, which promotes stem cell and progenitor cell proliferation [29]. Other soluble factors released from apoptotic cells include sphingosine-1-phosphate (S1P) [30] and endothelial monocyte-activating protein (EMAP) [31]^,^ processed by the caspase-dependent activation of SphK1/SphK2 and pro-EMAP respectively [24]. Together, chemo-attractants recruit myeloid-derived cells to alter cellular frequencies and extracellular remodeling of the stroma. In many cases, chemo-attractants alter immune cells’ polarization and/or the capacity of the immune cells to become activated. For example, depending on the expression of CD39 and CD73, which are often upregulated in the tumor microenvironment, ATP can be converted into adenosine, the latter tend to position and polarize macrophages into an M2 phenotype, leading to a vascular-endothelial-growth-factor-producing wound-healing-like phenotype in the tissue environment [32].

In addition to migration/chemoattractant factors, studies in a variety of invertebrate models (Hydra, Planaria, and Drosophila) as well as mammalian models have shown that apoptotic cells secrete soluble mitogens that physiologically influence cell growth in neighboring cells, a process termed apoptosis-induced proliferation (AiP), or cell-death-induced repopulation [33,34]. In Drosophila, elegant genetic screens whereby active caspases are expressed with a caspase inhibitor produced an “undead” phenotype; investigators identified a novel caspase-dependent role for Dronc1 (an initiator caspase) that resulted in the reactive oxygen species (ROS)/JNK-dependent secretion of mitogens such as Wingless (a WNT homolog), Dpp (a BMP/TGF-β homolog), and Spritz (an EGF homolog) that stimulated proliferation of neighboring cells [33]. Components of the Drosophila AiP pathway have been observed in mammalian hepatocytes and skin, demonstrating both initiator-caspase- and effector-caspase-dependent activation of ROS and JNK, as well as the secretion of mitogens such as Hedgehog [35] and PGE_2_ [36]. In the case of PGE_2_ production, activated caspases 3 and 7 cleave iPLA2 to generate arachidonic acid, which is further metabolized by Cox-2 and PGE_2_ synthesis, to promote proliferation and wound healing [37]. While AiP is arguably best characterized on epithelial cells [38], extracellular ROS gradients are also known to recruit and stimulate immune cells [39,40], which in turn can feedback and contribute to JNK-dependent signaling of epithelial cells [41]. Hence, despite the fact that regeneration in mammals is not as robust as in invertebrates, the AiP associated with caspase-dependent apoptosis appears evolutionarily conserved, and particularly exploited in cancer, whereby apoptosis mediates a compensatory response that includes the production of ROS and the secretion of mitogens that promote proliferation.

Apoptotic cells, via caspase activation, also secrete a series of extracellular membrane-delimited particles and vesicles that include exosomes (30–150 nm), ectosomes or microvesicles (100–1000 nm), and apoptotic bodies or blebs (1000 to 5000 nm) which can both indirectly influence the local microenvironment as well as make direct and long-range contact with neighboring cells [42]. Extracellular vesicles from apoptotic cells harbor find me signals that facilitate the directed migration of macrophages and phagocytes. For example, an active caspase proteolytically processed species of CX3C-chemokine ligand 1 (CX3CL1; also known as fractalkine) can associate on the outer membrane of apoptotic-derived microparticles to induce macrophage chemotaxis [43]. Similarly, caspase- and apoptosis-derived microparticles bearing ICAM-3 promote chemoattraction and tethering of apoptotic cells to macrophages [44]. Finally, in the case of apoptotic blebs, which occurs via caspase-dependent cleavage and activation of ROCK1 [45,46], blebs increase the membrane surface area apoptotic cells to facilitate interactions of immune cells with membrane-bound apoptotic materials, but also can stimulate macrophage chemoattraction [47]. Interestingly, apoptotic blebs also appear to package specific content, including the active relocation of chromatin and endoplasmic reticulum [48], as well as self-antigens that can be loaded into DCs [49]. Clearly, extracellular vesicles from apoptotic cells carry diverse cargo making them complex and dynamic entities. These vesicles include nucleic acids such as miRNA, lncRNA, and mRNA, as well as metabolites, lipids, and proteins that can induce the proliferation of neighboring cells and act as chemo-attractants in the tissue microenvironment, a term referred to as an “onco-regenerative niche” by Gregory and colleagues [50].

## 2. Physiological Signals from Efferocytosis in Normal Tissues

Concomitant with the production of soluble mitogen factors, chemo-attractants, and membrane-bound vesicles that harbor cell-extrinsic materials, apoptotic cells also directly mark their outer surface with recognition and “eat-me” signals that allow the dying cells to be clearly identified by phagocytes [51]. In many cases, eat-me signals are retained on the inner surface of the plasma membrane or (stressed) endoplasmic membrane in healthy cells, but translocated to the outer surface of the plasma membrane during apoptosis. Phosphatidylserine (PS), for example, is irreversibly translocated to the outer surface of the plasma membrane via the net activation of scramblases and inactivation of flippases in caspase-dependent mechanisms [52]. As such, during apoptosis, the combined activation of CED8/Xkr8 (a caspase-dependent PS scramblase) [53] and inactivation of ATP11C (a caspase-dependent flippase) [54,55] results in the irreversible localization of PS to the outer surface of the plasma membrane.

Subsequently, PS interacts with a cohort of diverse inhibitory PS receptors that recognize and/or internalize the apoptotic corpse. These receptors include direct PS-binding receptors comprised of T-cell immunoglobulin mucin family receptors (TIM-1, TIM-3, and TIM-4), stabilin, RAGE, and BAI-1, as well as indirect binding PS receptors that include Tyro3, Axl, and Mertk family receptors (TAM receptors) that are bridged to PS via Gas6 and Pros1 and b3/b5 integrins that are bridged to PS via MFG-E8 in a complex biology that is still not well-understood [2,5]. Part of the complexity of PS receptor biology and the inhibitory pathways that they regulate stems from the disparate classes of proteins that interact with PS (i.e., receptor tyrosine kinases (TAMs), G-protein-coupled receptors (BAI-1), integrins (MFG-E8/b5/b3 integrins), scavenger receptors (RAGE), and tethering receptors (TIMs)) and also from observations that PS receptors often collaborate to engage PS on the apoptotic cells and act cooperatively in at least a two-step process to modulate efferocytosis [9,56]. For example, in the case for efferocytosis in murine-bone-marrow-derived macrophages, TIM-4 and Mertk act as a functional PS receptor pair, whereby TIM-4 appears to tether PS-positive apoptotic cells to the surface of the phagocyte, thereby facilitating activation of Mertk as a signaling receptor (i.e., tyrosine phosphorylation and internalization of the apoptotic cell) [57,58]. Moreover, both TIM-4 [59,60] and Mertk [61] can also cooperate with avb5 and avb3 integrins via the PS opsonin MFG-E8 to functionally coordinate engulfment. These observations suggest at least a minimum two-step process for efferocytosis that involves a combined tethering step followed by an internalization step. However, it presently cannot be ruled for a more synergistic role for PS and apoptotic cell receptors that function as cohorts akin to what has been observed to the immunological synapse for T cell signaling [56]. While it is not exactly clear how different PS receptors functionally cooperate in post-receptor signaling, this arrangement clearly adds a complexity to the role of PS as a tolerogenic entity both under physiological signals as well as in cancer.

A second complexity in PS biology comes from findings that different cell types and efferocytes express different combinations of PS receptors, and that different PS receptors can be transcriptionally and post-translationally regulated to dynamically alter expression levels. In the case of resident alveolar macrophages (versus peritoneal myeloid-derived macrophages), alveolar macrophages express fewer TAM and TIM receptors such that alveolar macrophages are less efficient efferocytes than peritoneal macrophages [62,63]. Bone-marrow-derived macrophages express higher levels of Mertk compared to Axl [64], and, adding further complexity, Mertk and Axl are reciprocally regulated at the transcriptional level, whereby dexamethasone and anti-inflammatory cytokines, such as IL-10, induce Mertk and downregulate Axl, while LPS and poly-IC induce Axl and downregulate Mertk [65]. Moreover, Mertk is cleaved by ADAM17 under inflammatory conditions such as the addition of LPS, in patients with autoimmunity such as SLE, and in regions of atherosclerotic plaques, resulting in a non-functional Mertk receptor and reduced efferocytosis [66,67] As discussed below, this ability to dynamically alter inhibitory PS receptors depending on the cellular context and milieu of the tissue microenvironment has relevance in cancer models, particularly how these inhibitory signals can be therapeutically manipulated.

Finally, the above argument that different phagocytes have different combinations of PS receptors, as well as that PS receptors are subject to dynamic regulation by signals in the local microenvironments of the dying cells, also imply that not all efferocytic events may have the same post-receptor signaling and the same post-receptor tolerogenic and immunosuppressive outcomes. It is known for example, that the interaction of apoptotic cells with efferocytes (contact) is sufficient to inhibit LPS-inducible NF-kB activation since apoptotic cells retain this activity when phagocytes were treated with cytochalasin D to inhibit internalization and corpse degradation [68,69]. Likewise, mutation of Tyr867 in Mertk (an autophosphorylation docking site) can dissociate efferocytosis from the ability of Mertk to inhibit TLR4 signaling to NF-kB showing the efferocytosis and inhibition of NF-kB can be mediated by distinct steps [70]. By contrast, interesting studies by the Ravichandran and Tabas laboratories have shown that mitochondrial dynamics—including the expression and the UCP2 uncoupler protein that controls mitochondrial membrane potential [71] as well as mitochondrial fission (via dynamin-related protein 1) [72]—are required for continued efferocytosis. In the case of mitochondrial fission, these studies showed that mitochondrial fission allows for an increase in cellular calcium and subsequent phagosome maturation and the tolerogenic signals associated with efferocytosis [72]. Finally, recent studies showed that in LC3-associated phagocytosis (LAP), in which autophagy proteins are involved apoptotic cell clearance, phagosome formation and maturation is required for tolerogenic signals such as IL-10 production and immune silencing [73,74]. Indeed, a common theme in PS biology and PS-dependent efferocytosis is that final degradation of the corpses is generally immunosuppressive and tolerogenic, and physiological efferocytosis has been shown to suppress the production of pro-inflammatory factors, such as TNF-α; induce the production of immunosuppressive factors, such as IL-10; and induce pro-resolving factors and lipoxins, as well as inhibit the generation of ROS and NF-kB. However, while clues are emerging that different types of dying cells can traffic to different intracellular itineraries, it is still not completely clear how the binding and trafficking of apoptotic cells leads to specific tolerogenic signals and in what compartments these signals emerge. Clearly, physiological apoptosis coupled with efficient efferocytosis, protects tissue function, and contributes to a myriad of regenerative and immunosuppressive events that control homeostasis.

## 3. The Dark Side of Apoptosis and Efferocytosis in Cancer Progression

The association of apoptosis and efferocytosis with regeneration signals, wound healing, and tolerogenic immune signals have relevance in cancer biology. Although resistance to apoptosis is a common feature of cancers, cancer cells are not fully resistant to apoptosis. Support has been demonstrated for an extension of AiP into clinical models of cancer. The presence of apoptosis in the tumor microenvironment is demonstrated as an “apoptotic index” (AI), defined as the number of apoptotic cells per 100 intact neoplastic nuclei. Consistent with the idea that apoptotic cells can induce AiP, often the AI is correlated with the proliferative index, as a measure of the percentage of Ki-67-expressing cells. Under physiological conditions, AIs are estimated in the range of 0.1% to 2.0%, even in highly proliferative tissues such as the immature thymus, in B cell germinal centers, and in testes [21]. However, in contrast, it has been reported that with increasing markers of tumor aggressiveness, AIs in tumors can reach 5–15% or higher, where AI correlates with the malignant features of tumors that include metastasis, mortality, and recurrence following therapeutics [75,76,77]. Indeed, as Ucker and colleagues articulate [21], there is emerging evidence that the AI serves as a leading prognostic marker and independent predictor for overall survival or disease-free survival. This stems from an understanding of apoptotic signaling in healthy tissues, such as wound regeneration, and the analogy of tumors as “wounds that do not heal” [78,79]. Cells neighboring a site of injury where apoptosis occurs may undergo sustained compensatory proliferation in a finite time until the wound heals. However, in tumorigenesis, including recurrence after treatment with chemotherapeutics and radiotherapy, sustained compensatory proliferation over weeks and months is likely to promote accelerated repopulation providing an explanation for this paradoxical phenomenon as well as a hindrance to cancer therapeutics. Although these events are likely to be complex and multifactorial, interesting studies by Huang and colleagues showed that caspase activation in 4T1 breast tumors led to caspase-dependent iPLA2 cleavage PGE_2_-mediated tumor repopulation (a hallmark of apoptosis-inducible proliferation), and decreased tumorigenicity was observed in either caspase 3 or iPLA2 knockout cells [22]. In radiation models, focal radiation induces an increase in PKC zeta [80] activity and increased proliferation, suggesting that specific anti-cancer agents might produce unique effectors of AiP.

At a genetic level, high apoptotic indexes may likely be dictated by oncogenes that regulate apoptosis, and are clearly influenced by tumor heterogeneity. For example, in non-Hodgkin’s lymphoma (NHL) and melanoma, a non-intuitive link between low expression of Bcl-2 and a high AI has been associated with increased mortality, suggesting that a high AI may be independently prognostic [81]. These studies also addressed a novel parameter that correlated the mitotic and apoptotic indices into the “turnover index” (TI). Such results indicate that a lower TI score was correlated with longer overall survival than those NHL cells in the high TI groups, suggesting a prognostic relationship. Additional evidence exists to explain the paradoxical observation that higher Bcl-2 expression corresponds to a better prognosis in some individuals, suggesting a vacant niche filled by higher levels of apoptosis leading to tumorigenesis or dysregulation of proliferation during tumor growth [81]. Such events provide a paradox in cancer biology given the evasion of apoptosis, considered a general hallmark of cancer phenotypes, and the supreme goal of targeted therapy is to induce apoptosis [82]. Unfortunately, it appears that unabated apoptosis can have an unanticipated consequence to promote tumor progression, and particularly targeting apoptosis proteins (such as Bcl-2, Bcl-xL, and PI3-kinase) should be considered with adjuvant immunogenic therapeutics.

In addition to AI and compensatory proliferation and cancer repopulation, increased AI also necessitates an increased demand for efferocytosis that indirectly drives tolerogenic signals in the tumor microenvironment [52]. As such, many solid cancers display elevated externalized PS, an event that renders the tumor microenvironment immunosuppressive [52]. High externalized PS in the TME constitutively engages PS receptors such as TAMs and TIMs on infiltrating myeloid cells and T cells, suppressing the ability to engage host anti-tumor immunity [83]. In the following sections, we address approaches to manipulate apoptosis in the tumor microenvironment, as well as approaches to manipulate efferocytosis.

## 4. Shifting Classical Apoptosis towards Immunogenic Cell Death and Non-Caspase-Mediated Forms of Cell Death

Given the high levels of cell death in the tumor microenvironment, and the potential influence of apoptotic cells on tumor progression, therapeutic and experimental strategies that regulate the extent of caspase activation and/or the extent by which apoptotic cells are cleared could manage the context of dying cells in tumor progression (Figure 2). One approach to alter the anti-tumor immunity is to alter apoptosis to trigger an immunogenic cell death pathway via specific cell death inducers. Elegant work pioneered by Kroemer and colleagues observed that certain chemotherapies such as doxorubicin (an anthracycline), certain radiotherapy approaches (focal high dose ionizing radiation), and some physiochemical therapies (hypericin-based photodynamic therapy) could augment tumor immunity through a caspase-dependent immunogenic form of cell death that ultimately lead to DC- and CD8-mediated T cell-mediated immunity [84,85,86]. Mechanistically, anthracyclines and other induces of immunogenic cell death promote the exposure of dominantly acting secreted and cell surface molecules that signal danger instead of immunotolerance. One of the principal components of immunogenic cell death appears to be the exposure of a specific eat-me signal, calreticulin (CRT), on the cell surface [87,88]. Calreticulin, normally present in the endoplasmic reticulum lumen, translocates to the extracellular surface of cancer cells during endoplasmic reticulum stress and ICD, acting as a prominent eat-me pro-phagocytic signal by interacting with CD91 on the surface of the DC [89]. Notably, blocking CRT exposure on anthracycline-treated tumor cells can markedly reduce DC-mediated cross-presentation and anti-tumor immunity [90,91,92] while supplying endogenous CRT to dying tumor cells that cannot display endogenous CRT enhanced their immunogenicity [93], suggesting that CRT acts dominantly, over PS-mediated immunosuppression, and appears to be capable to unmask aforementioned tolerogenic signals on apoptotic cells.

However, besides CRT, ICD also leads to the release of a number of additional DAMPs that have complex pleotropic effects that lead to therapeutically pertinent adaptive tumor immunity (for excellent reviews see reference [94,95]). These include, but are likely not limited to, (i) high mobility group 1 (HMGB1) that activates TLR4 and a pro-inflammatory pathway shown to be important for cross-presentation of tumor antigens to DCs, bypassing the lysosomal degradation of engulfed tumor antigens necessary for cross-presentation [96,97]; (ii) secreted ATP which as noted above acts as a chemo-attractant factor but also an activator of the inflammasome [98]; (iii) active secretion of type I IFNs [99]; (iv) heat shock proteins (HSPs), such as HSP70 and HSP90 [95]; and (v) the passive release of nucleic acids that can activate TLRs [100]. Interestingly, recent studies indicate that the PS-binding protein annexin A1 (ANXA1) [101,102] is also released during ICD and has a critical role in establishing anti-tumor immunity [103]. Since PS is also externalized during ICD, it is possible that ANXA1, in part, naturally binds/blocks PS on ICD dying cells in addition to its ability to bind G protein-coupled formyl receptors 1 and 2 (FPR1 and FPR2), and appears to override tolerogenic signals of PS.

## 5. Non-Caspase-Mediated Alternative Forms of Cell Death

In addition to immunogenic cell death, whereby caspases are activated concomitantly by the intrinsic apoptosome pathway, as well as release danger and immunogenic constituents that act dominantly, other forms of cell death that include necroptosis, pyroptosis, and ferroptosis achieve cell demise independently of classical executioner caspase 3 and 7 activation and, in principle, spare the tumor microenvironment the post-caspase and the pro-tumorigenic events described above. Necroptosis, a caspase-independent form of cell death characterized by RIPK1 and RIPK3 activation and phosphorylation of MLKL to induce necrosome formation, produces a lytic type of immunogenic cell death that can release DAMPs and elicit robust inflammation as well as adaptive immune responses against tumor cells by activating DCs [104,105]. Since necroptotic cell death is generally activated under conditions where caspases are inhibited, for example under conditions where cFLIP/caspase 8 heterodimers prevent caspase 8 activation [106], or when caspase 8 is pharmacologically inhibited or mutated [107,108], it is predicted that necroptosis may spare tumors of the pro-tumorigenic role of caspases articulated above by supporting RIPK3-mediated necroptosis. Recent studies by Ning and colleagues have shown that activate caspase (3 and 7) can cleave an inactivate cGAS, MAV, and IRF3 to suppress type I IFNs, thereby preventing important immunogenic effector functions [109]. In other models of chronic hepatic inflammation (a tissue that is well-suited to regenerate), the activation of RIPK3 in hepatocytes reduces compensatory proliferation in nonparenchymal liver cells by blocking caspase-8-dependent activation of JNK [110,111].

In relation to tumor immunity, analogous to the events associated with ICD that release DAMPs associated with intracellular components, necroptosis, being a lytic phenotype, also exposes tumor-associated antigens and tumor-specific antigens to DCs, cross-priming them to mount a pro-inflammatory adaptive response, i.e., attack by cytotoxic CD8 lymphocytes. Studies by Yatim and colleagues [112] and Aaes and colleagues [113] showed that RIPK1 and NF-kB signaling can mediate cross-priming of CD8^+^ T cells and that vaccination with necroptotic cells induces efficient anti-tumor immunity, respectively (the latter the gold standard for immunogenic death). In a more recent study by Snyder and colleagues, intra-tumoral injections of necroptotic cells (by engineering constitutively active forms of RIPK3), but not apoptotic cells, into B16 tumors induced BATF3 cDC1 and CD8^+^ T cell-dependent anti-tumor immunity via the RIP1/RIP3/NF-κB pathway independently of cell lysis [114]. These data demonstrated that modulating death machinery towards necroptotic death in conjunction with immune checkpoint inhibitors in a tumor microenvironment induces durable tumor clearance. Inducing necroptotic death may be particularly important in the treatment of “cold tumors” (i.e., immunologically silent due in part to the defective recruitment of APCs and lack of T cell activation/homing) where there is a lack of exposure of tumor antigens. By inducing lysis, necroptosis is more likely to expose tumor antigens and priming of the tumor microenvironment to be conducive to cross-presentation and T cell recruitment.

In recent years, other inflammatory forms of cell death, including pyroptosis and ferroptosis, have been described in the context of tumor immunity, characterized both specific inducers as well as the release of DAMPs, such as HMBG1, as well as specific immunogenic components unique to these death modalities [115]. In the case of pyroptosis, characterized by cell swelling, lysis, and the release of several pro-inflammatory factors that specifically include IL-1b and IL-18, the formation of inflammasome (NLPD3/caspase 1) and the eventual lytic cell death by the release of a pore-forming protein called gasdermin [116]. While recent studies indicate that exogenous activation of pyroptosis can elicit strong anti-tumor activity [117,118], possibly by shifting from caspase-3-dependent apoptosis [119], definitive evidence that pyroptosis induces a classic ICD will require a prophylactic tumor vaccination model to access whether the DAMPs associated pyroptotic death are sufficient for tumor priming.

Likewise, ferroptosis, a form of cell death characterized by iron-dependent lipid peroxidation from polyunsaturated fatty acid (PUFA) sidechains of membrane phospholipids (PLs) become oxidized due, in part, to cellular depletion of GSH or inactivation of GPX4, the latter binds to and detoxifies membrane-bound peroxide phospholipids [94]. The presence of redox-active iron, the peroxidation of PUFA-PLs, and inhibition of lipid peroxide repair mechanism constitute the set of characteristic conditions associated with ferroptosis, including the potential for the oxidation of catalytic cysteines on active caspases. Such autocatalytic buildup of reactive oxygen species (ROS) and peroxidation of membranes is associated with specific DAMPs during ferroptosis, and, like pyroptosis, likely has complex interplay to impinge on tumor immunity. Recent studies by Wang and colleagues showed mechanistically that interferon gamma (IFN-γ) released from CD8^+^ T cells downregulates expression of SLC3A2 and SLC7A11, subunits of the glutamate–cystine antiporter required for cystine uptake, and, as a consequence, promotes tumor cell lipid peroxidation and ferroptosis [120,121]. Similar to pyroptosis, it will be interesting to assess with prophylactic vaccination studies whether ferroptosis can be induced with ectopic inducers to address whether ferroptosis acts in an ICD-like manner that can cross-present tumor antigens.

Clearly, while the conceptual idea of manipulating apoptosis towards ICD and non-caspase-mediated forms of cell death is clearly meritorious, challenges arise in the above-mentioned discussions in complex tumors, since many investigations in murine models are performed with homogeneous vaccination clonal models that do not reflect genetic heterogeneity in complex spontaneous cancers. Indeed, given the genetic and physical heterogeneity of solid cancers [122,123], it is likely that a range of different types of cell death will occur simultaneously in complex tumors (Figure 3), and that caspase-mediated events will be dynamic in the tumor microenvironment [50]. Issues of genetics, epigenetics, hypoxia, ATP levels, and autophagy (as a survival mechanism that maintains ATP) will impact the mode of cell death and level of immunogenicity. Factors include differences in genetics, epigenetics, and mutational burden across solid tumors, as well as genes and networks associated with expression of caspases, RIP3K and RIPK1, inflammasomes, and redox status of the tumor cell. As noted above, suppression of caspases can prevent the degradation of genes that regulate type I IFNs [109], shift death towards a necroptotic pathway, and influence the distribution of immunogenic responses in the tumor microenvironment.

In addition to tumor heterogeneity at the genetic level, apoptotic versus caspase-independent cell death in complex cancers are influenced by the extent of hypoxia, metabolic stress, altered neo-angiogenesis (including the formation of leaky capillaries with altered vascular permeability towards clotting factors and immune cells), and autophagy. Since both apoptosis and pyroptosis depend on ATP for the formation of the apoptosome and inflammasome, respectively, it is expected that areas of poorly vascularized tumors and central cores will favor necroptotic and ferroptotic death pathways while vascularized margins would favor apoptosis. It is likely the prevailing form of cell death within specific areas of tumors will be regulated by complex shifts and gradients in oxygen concentration, nutrient availability, leukocyte penetration, and adaptability of the cancer cells to the changing environment. For example, the rapid growth of tumors and increased expression of angiogenic factors such as vascular endothelial growth factor (VEGF) and angiopoietin (ANGPT1) can destabilize the tumor vasculature, causing improper formation and hemorrhaging of newly formed blood vessels within the tumor microenvironment leading to poor delivery of oxygen and hypoxic conditions [124,125]. Oxygen diffusion within tumors varies greatly depending on the stage, size, and location of the tumor. Within the center or core of the tumor, poor vascularization prevents oxygen and glucose from reaching these cells leading to hypoxia and depletion of ATP. Recent studies also show that in pancreatic ductal adenocarcinoma (PDAC) MHC-I molecules are selectively targeted for lysosomal degradation by an autophagy-dependent mechanism that involves the autophagy cargo receptor NBR1 [126]. Therefore, hypoxic autophagy could have an unanticipated consequence to drive immune escape in the tumor microenvironment.

## 6. Modulation of PS Signaling and Efferocytosis as an Anti-Tumor Strategy

Given the high level of apoptosis and PS in tumors, another attractive idea is to counteract the immunosuppressive effects of apoptotic cells with PS blocking agents (Figure 4). Early studies by Herrmann and colleagues showed that interfering with PS-mediated efferocytosis by masking PS with annexin V favors an anti-tumor host response, possibly by delaying clearance and possibly promoting a sterile inflammation by the release of DAMPs by secondary necrotic death (i.e., ICD mimicry) [127,128,129]. More recently, a series of PS-targeting mAbs and PS-targeting biomolecules have been developed that can interfere with PS-mediated immunosuppression and facilitate the induction of an innate and adaptive anti-tumor immune response (reviewed in references [52,130,131]). While it might be expected that PS-targeting mAbs simply bind to and mask PS on the surface of apoptotic (and other PS-positive) cells to disrupt immunosuppressive signals from PS receptors such as TAMs and TIMs, emerging research identifies a more complex biological function for PS-targeting mAbs. In the case for bavituximab (Bavi), a mouse–human chimeric IgG1 antibody, which has been characterized for immuno-oncology applications in mouse preclinical studies and in human phase 2 and phase 3 studies, binds PS indirectly via a serum cofactor b2-glycoprotein (b2GPI), a direct PS binding protein.

Mechanistically, although still not completely understood, Bavi has a proposed complex mode of action. Binding of Bavi to b2GPI induces a conformational change to b2GPI and stabilizes a dimeric complex of b2GPI to PS on both apoptotic cells and viable PS-positive cells [132]. Subsequently, the Bavi/b2GPI/PS complex appears to have multifunctional activities that include (i) partial masking of PS and blockage of PS receptors such as TAMs and TIMs, (ii) induction of ADCC on viable tumor vascular endothelial cells, (iii) polarization of M2 toward M1 macrophages, and (iv) maturation of DCs towards an active cross-presenting phenotype that activates host anti-tumor immunity [130].

To date, Bavi has cleared Phase I as well as Phase II clinical trials and demonstrated an acceptable safety profile. In 2013, Bavi was being evaluated in SUNRISE (**S**timulating Imm**U**ne Respo**N**se th**R**ough Bav**I**tuximab in a Pha**SE** III Lung Cancer Study), a global, randomized, double-blind, placebo-controlled registration trial sponsored by Peregrine Pharmaceuticals. The clinical trial was to assess Bavi plus docetaxel versus placebo plus docetaxel in 582 patients with previously treated locally advanced or metastatic non-small cell lung cancer NSCLC (NCT01999673). The clinical trial was unfortunately terminated because the combination of Bavi and docetaxel did not exhibit beneficial overall survival versus the docetaxel alone [133] (median overall survival (OS) was 10.5 in the docetaxel + Bavi combination and 10.9 months in the docetaxel + placebo). There were no differences found in progression-free survival and toxicities were minimal in each intervention. Although this was a disappointing outcome, the one key takeaway from the study showed that patients who had elevated levels of serum β2GPI exhibited a non-significant OS trend which favored the Bavi + docetaxel arm (hazard ratio (HR): 1.06; 95% confidence interval (CI) 0.82–1.22; *p* = 0.134). This indicated that increased levels of β2GPI would enhance the efficacy of Bavi by increasing the binding sites to PS in the TME.

Based on previous preclinical observations that PS-targeting mAbs can activate T cell-mediated immunity, this targeting strategy may also have therapeutic potential as combinatorial strategies with conventional checkpoint therapeutics such as anti-PD1 and anti-CTLA4 [134,135]. Indeed, follow up analysis from patients previously enrolled in SUNRISE and received post-study immune checkpoint inhibitor therapy, OS favored the Bavi + docetaxel arm (HR 0.46; 95% CI 0.26–0.81; *p* = 0.006), versus docetaxel alone, suggesting that Bavi treatments altered the TME in a way that allowed for a better response to immunotherapy. Furthermore, analysis of circulating cytokines in these patients demonstrated that low pretreatment serum levels of IFN-γ associated with better activity of Bavi + docetaxel [136], indicating that Bavi may increase the priming of T cells and that the combination of PS targeting mAbs plus immunotherapy might trigger an ICD-like immune response.

Indeed, there is precedent to indicate that Bavi combination with immunotherapy is an effective approach to cancer. The first was PS-targeting antibody 1N11 was found to synergize with anti-PD-1 immunotherapy and exhibit anti-tumor immunity in a murine model of triple-negative breast cancer. Using two breast cancer models, EMT-6 and E0771, in immunocompetent mice, 1N11 was administered as a monotherapy or in combination with anti-PD-1 [135]. 1N11 treatment alone was found to inhibit tumor growth and also enhance the anti-tumor effects of anti-PD-1 therapy including increasing the levels of infiltrating lymphocytes into the TME. In a separate study, Freimark and colleagues demonstrated that the combination of anti-CTLA-4 or anti-PD-1 immunotherapies with PS-targeting agent 1N11 synergized and exhibited anti-tumor properties in a mouse model of melanoma [134]. Within these studies, the authors demonstrated that the combination enhanced tumor-infiltrating CD4 and CD8 cells, along with increased levels of pro-inflammatory cytokines. Additionally, the combination also resulted in the increase of CD8 T to myeloid-derived suppressor cell (MDSC) ratio within TMEs, indicating a pro-inflammatory shift in the immune milieu. These data together provide strong preclinical evidence to combine PS-targeting with immunotherapy in cancer. Recently, Oncologie Inc. (current owner of Bavi) has announced two new clinical trials that are now recruiting and involve a combinatorial intervention of Bavi and anti-PD-1 (KEYTRUDA, Merck): Phase II Open Label Study in Advanced Gastric and GEJ Cancer Patients (NCT04099641) and Phase II Open Label Study in Advanced Hepatocellular Carcinoma (NCT03519997). The outcomes of the Bavi trials, as well as future studies developing novel PS-targeting molecules, such as the PS-binding peptide–peptoid hybrid, PPS1D1 [137]; PS-targeting nanovesicles (SapC-DOPS) [138,139]; and bispecific antibodies will be necessary to assess whether PS-targeting approaches will have clinical utility in immuno-oncology.

## 7. Targeting PS Receptors in Immuno-Oncology

An emerging and complementary strategy to the targeting of PS described above using PS-targeting mAbs that is showing therapeutic promise in IO involves the targeting and inhibition of certain PS receptors, most notably Mertk and TIM-3 expressed on tumor-associated macrophages and/or on T cells. In the case for TAMs (Tyro3, Axl, and Metk), while these receptors can be expressed and upregulated on tumor cells to drive proliferation, survival, EMT, and metastasis [140], they are also expressed on immune cells that generally transmit inhibitory signals for TLRs, inflammasome, and IFNs [17,141]. For example, Axl on DCs and macrophages, when activated by its ligand Gas6, results in the upregulation of negative TLR and cytokine regulators, suppression of cytokine signaling 3, and suppression of cytokine signaling 1, dampening immune activation [3]. Moreover, interesting studies by Rothlin and colleagues have shown that Axl and Mertk, expressed on macrophages, are required for the up-regulation of IL-4 and IL-13 and for subsequent tissue repair [17]. These studies showed sensing of IL-4 in the presence of apoptotic cells promotes the expression of key tissue repair factors in macrophages. Neither signal on its own is sufficient to induce this program, which included the upregulation of *Arg1*, *Retnla*, and *Chil3* [16]. Interestingly, macrophage IL-4 also results in the activation of invariant natural killer T cells (iNKT), resulting in the resolution of inflammation [142]. Finally, in addition to macrophages, TAM receptors (Mertk) appear to be important for NK cells as well, as inhibitors of Mertk enhanced anti-metastatic NK activity in vivo [143].

In the case of Mertk, which is highly expressed on peripheral macrophages as well as tumor-associated macrophages, its expression correlated with poorer survival outcomes. Both genetic and pharmacological strategies to block Mertk function dampen host anti-tumor immunity [144,145,146,147]. For example, interesting studies by Crittenden and colleagues have shown that knockout of Mertk in macrophages improves radiotherapy outcomes [148], while more recent studies using anti-Mertk neutralizing antibodies have shown that blocking Mertk-dependent efferocytosis induces a secondary necrosis of tumor cells that release cGAMP [149]. Extracellular cGAMP, in turn, is internalized in macrophages and activates STING-dependent transcriptional-dependent up-regulation of IFN-β that enhances host anti-tumor immunity [149]. Similar results have been observed in radiotherapy models when anti-Mertk mAbs were combined with anti-PD1 mAbs [150]. Independent studies have also shown that blocking Mertk on macrophages suppresses efferocytosis and can synergize with anti-PD1 therapeutics, potentially shifting efferocytosis from tumor-associated macrophages into a DC cross-presentation competent compartment (Davra, in press). Together, these data suggest that Mertk is an important checkpoint on tumor-associated macrophages that impinges on efferocytosis, and can complement chemotherapies and radiotherapies. Such a blockage of efferocytosis and alteration in the processing to dying cells would provide an alternative mechanism to what has been observed with anti-CD47, a don’t-eat-me signal on the surface of live cells, and the development of anti-CD47 therapeutics [151,152,153,154,155,156]. However, while the rationale for blocking PS and PS-R (TAMs) as inhibitory signals in cancer is clearly supported [64,157], in certain inflammatory cancers, blocking such pathways may have unanticipated consequences. For example, azoxymethane (AOM)- and dextran sulfate sodium (DSS)-induced inflammation-associated cancer was shown to be exacerbated in mice lacking Axl and Mertk [158]. In this latter model, the ablation of Axl and Mertk signaling is associated with increased production of proinflammatory cytokines and failure to clear apoptotic neutrophils in the intestinal lamina propria, thereby favoring a tumor-promoting environment [158].

In addition to the TAM receptors, T-cell transmembrane immunoglobulin and mucin receptors (TIM receptors) also comprise a family of PS receptors (TIM-1, TIM-3, and TIM-4) that are expressed on both tumor cells and immune cells, and, like TAM receptors, TIM receptors appear to act as inhibitory receptors that dampen host tumor immunity. In the case for TIM-3, that is expressed on both DCs and T cells, expression of TIM-3 is associated with both the inhibition of antigen presentation on DCs and on exhaustion of T cells, and TIM3 antibody treatment reverses these phenotypes and improves tumor outcomes [131]. While it is not yet clear whether TIM-3 has a role in modifying efferocytosis, previous studies have shown that TIM-4 cooperates with Mertk in driving efferocytosis [58].

## 8. PS Exposure Occurs on Viable Activated Cells and During Non-Apoptotic Forms of Cell Death

While the aforementioned studies focused on the role of PS targeting and PS-R targeting mAbs on apoptotic cells, and blocking tolerogenic signals from apoptotic cells, it should be noted that viable stressed cells and live activated immune cells externalize PS in the tumor microenvironment. Mechanistically, the externalization of PS on viable cells is clearly distinct from irreversible PS on apoptotic cells which is mediated by the calcium-activated phospholipid scramblase TMEM16F [159]. While externalized PS does not appear to result in the engulfment of viable cells [160], suggesting that the topology and/or density of PS is not sufficient as an eat-me signal, calcium-mediated PS exposure is sufficient to activate PS receptors (TAMs) [161] and likely drive immunosuppressive and tolerogenic signals. Constitutive PS exposure on viable vascular tumor endothelial cells can be targeted by PS-targeting antibodies (Bavi) and thought to mechanistically contribute to ADCC [130,162,163], and studies by Rothlin and colleagues showed that externalized PS on activated T cells serves as an exhaustion signal, as a negative feedback to immune activation [164]. Such externalized PS can recruit PS-binding proteins such as Gas6 and Pros1, and negatively feedback to inhibit TAM receptors and antigen presentation on DCs. Finally, tumor-associated exosomes, originating from live tumor cells, have been shown to be PS-positive and possible biomarkers for solid cancers and tolerance [165,166]. Therefore, PS/PS-R-targeting strategies are expected to have pleotropic activities, and may support tumor immunity by multiple activities.

While the above arguments support the emerging conceptual idea that constitutively exposed PS on apoptotic and viable stressed cells in the tumor microenvironment can be therapeutically targeted to improve tumor immunity, a caveat of this idea is the recent observations that cells dying by necrosis and other forms of regulated necrosis such as necroptosis, ferroptosis, and pyroptosis have been shown to express PS, albeit without the caspase-mediated phospholipid scramblases and flippases responsible for PS flipping during apoptosis [167,168]. In the case of necroptosis, recent evidence indicates a TMEM16F-independent mechanism whereby PS externalization occurs prior to membrane rupture and requires the activation of MLKL and components of the ESCRT-III pathway [169]. It is not yet clear whether PS externalized by caspase-independent mechanisms is functionally distinct from caspase-dependent PS, for example, whether the topology, density, and oxidative state are different, nor is it clear whether such PS activates PS receptors such as TAMs and TIMs to a similar capacity. It is likely that components of ICD and DAMPs for caspase-independent death act dominantly to classical PS-mediated tolerance, although it will be of interest to determine if PS targeting agents augment ICD by enhancing DAMP signaling.

## 9. Conclusions

The physiological fate of cells that die by apoptosis is their rapid clearance by efferocytosis. There is accumulating evidence that these events, if not properly controlled, can paradoxically be mediators of tumor progression and immune escape. Targeting these pathways to control the extent of apoptosis-induced proliferation and efferocytosis-mediated immune escape will attempt to harness these pathways into novel cancer therapeutics and modalities. Future preclinical and clinical studies that combine immunogenic strategies with the blockade of tolerogenic signals should provide novel approaches in immuno-oncology.

## Figures and Tables

**Figure 1 cells-09-02207-f001:**
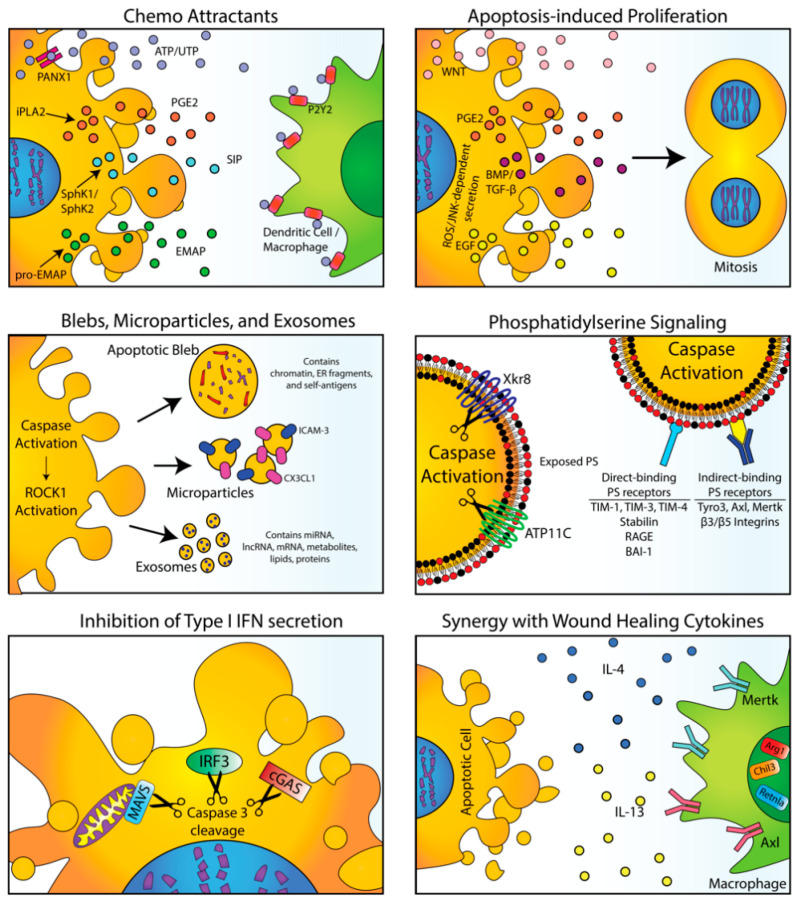
Caspase-dependent interactions between apoptotic and viable cells. Top left: Chemo attractants. Signaling molecules (including ATP/UTP, PGE2, S1P, and EMAP are released by apoptotic cells and attract immune cells (including dendritic cells and macrophages). Top right: Apoptosis-induced proliferation. Released signaling molecules (WNT, PGE2, BMP/TGF-β, and EGF) promote the proliferation of nearby cells. Bottom left: Blebs, microparticles, and exosomes. Apoptotic cells release vesicles of various sizes containing various signaling molecules taken from the dying cell. Bottom right: Phosphatidylserine (PS) signaling. PS is exposed on the outer surface of apoptotic cells by the activation of scramblases (i.e., Xkr8) and the inactivation of flippases (i.e., ATP11C). PS exposure interacts with direct-binding PS receptors (that may act as tethering receptors) and indirect-binding PS receptors, which require PS-sensing ligands and are known to promote immunosuppressive signaling and efferocytosis.

**Figure 2 cells-09-02207-f002:**
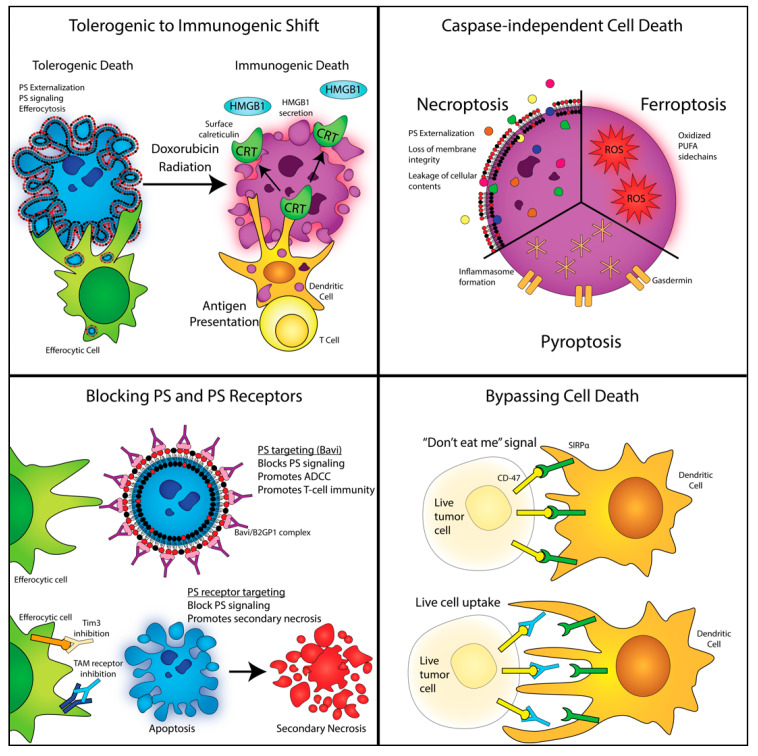
Targeting apoptotic and efferocytic pathways in cancer immunology**.** Top left: Tolerogenic to immunogenic shift. Shifting to an immunogenic cell death via chemotherapy and radiation therapy promotes the influx of antigen-presenting cells, whereby T cells can become educated against the tumor. Top right: Caspase-independent cell death. Caspase-independent cell death promotes immunogenicity via multiple mechanisms. Though phosphatidylserine (PS) is externalized on necroptotic cells, the loss of membrane integrity and release of intracellular contents “override” any immunosuppressive signaling. Ferroptosis and pyroptosis promote an inflammatory response by either oxidation or the formation of gasdermin pores, respectively. Bottom left: Blocking PS and PS receptors. Blocking PS signaling, either by targeting PS directly or PS receptors, prevents efferocytosis and leads to secondary necrosis, promoting an inflammatory response. Bottom right: Bypassing cell death. Targeting CD-47/SIRPa, the “don’t-eat-me” signal, permits the uptake of live tumor cells. Note: HMGB1—high mobility group 1; CRT—calreticulin; PUFA—polyunsaturated fatty acid; TAM—Tyro3, Axl, and Mertk; TIM—T-cell immunoglobulin mucin (TIM); Bavi (Bavituximab).

**Figure 3 cells-09-02207-f003:**
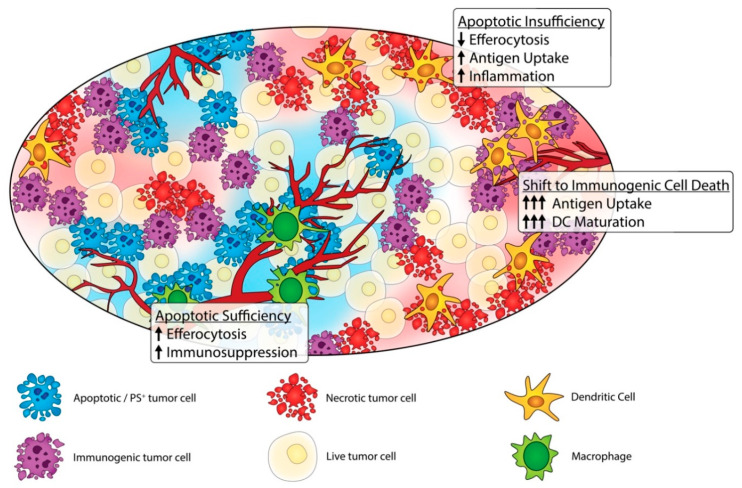
Heterogeneity of cell death in solid tumors. Immunosuppressive, apoptotic cells (blue) promote efferocytosis and immunosuppression, potentially promoting tumor growth. Conversely, immunogenic (purple) and necrotic (red) cells attract more antigen-presenting cells, promote antigen uptake, and encourage dendritic cell maturation resulting in a more robust anti-tumor response.

**Figure 4 cells-09-02207-f004:**
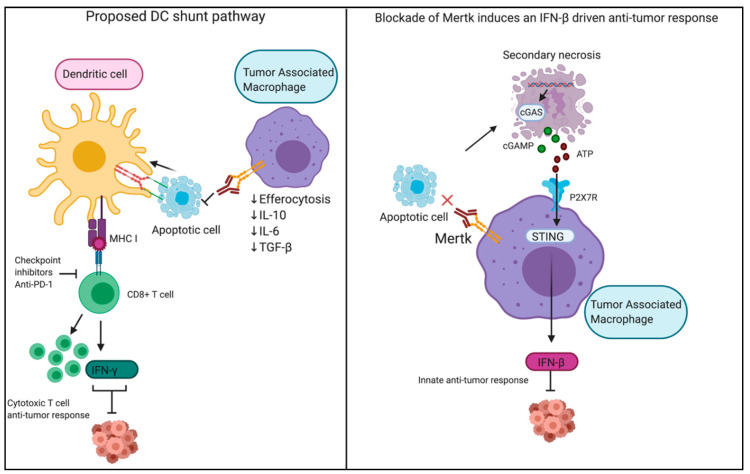
Inhibition of the efferocytosis receptor Mertk drives anti-tumor responses. In the right panel, the mechanism described by studies done by Zhou et al. is depicted wherein the blockade of Mertk by a monoclonal antibody induces an IFN-β driven anti-tumor response. Uncleared apoptotic tumor cells undergo secondary necrosis and release cGAMP, which enters and activates STING in tumor-associated macrophages, activating a local type I IFN response and IFN-mediated enhanced T cell immunity. We further propose a model of enhanced antigen cross-presentation by Axl-expressing DCs which could efferocytose apoptotic cells under the blockade of Mertk-driven efferocytosis by macrophages. Created with BioRender.com.

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
