# Peer review of "Cell Death in the Tumor Microenvironment: Implications for Cancer Immunotherapy"

_cells, 2020, doi:10.3390/cells9102207_

Round 1

Reviewer 1 Report

In this review article entitled “Cell death in the Tumor Microenvironment: Implications for cancer Immunotherapy”, the authors reviewed current knowledge on various forms of cell death that have impacts on cancer immune escape and progression in tumor microenvironment and how immunogenic cell death can augment host immunity against cancer. Importantly, the authors described the bench-to-bedside translational research to develop cancer immunotherapeutic approaches by targeting tolerogenic mediators derived from the events in apoptosis, efferocytosis, and immunogenic cell death. Overall, this is a comprehensive and thorough review on how cell death events in tumor microenvironment impacts on tumor immunity and growth. Some comments are listed below:

  1. Does apoptosis-induced proliferation in tumor microenvironment promotes growth of all neighboring cells, including tumor and immune cells?
  2. Elaboration of similarities and differences of chemoattractants-mediated immune tolerance generated from normal physiological apoptosis, tumor apoptosis and live cancer cells should help understand the establishment of tumor resistance of host immunity.
  3. Anything known about differences in compensatory proliferative signals generated after radiation/chemo-drug treatments which induce massive death vs compensatory proliferative signals generated by spontaneous tumor cell death?
  4. Complexity of cell-cell interactions makes the figure 2 difficult to read.
  5. Check the spelling of all the text fields. There are some typos, for examples: “consititive” (line 541) and “cleanance” (line 565).

Author Response

Reviewer #1 had generally supportive comments that “this is a comprehensive and thorough review on how cell death events in the tumor microenvironment impacts tumor immunity and growth”.   The reviewer also articulated several comments and queries that might be addressed.

Point #1: Does apoptosis-induced proliferation in tumor microenvironment promotes growth of all neighboring cells, including tumor and immune cells?

Reply:  This is an interesting point concerning the complexity of AiP in the tissue microenvironment. While AiP is best characterized for epithelial cells (Fogarty and Bergmann, CDD 2017), there appear to be several context and tissue-specific forms of AiP.  For example, in many systems, extracellular ROS gradients are known to recruit and stimulate immune cells (Neithammer et al, Nature 2009; Razzell et al (Curr Biol, 2013), which in turn can in turn feedback and contribute to JNK dependent signaling of epithelial cells (Fogarty et al, Curr Biol 2016).  In the revised MS, we have added sentences on p3 (lines 109 (plus the aforementioned references) that while AiP is best characterized for epithelial cells, it is likely to impact other cell types including progenitor and immune cells that contribute to the regenerative response.

Point #2:  Elaboration of similarities and differences of chemo-attractants-mediated immune tolerance generated from normal physiological apoptosis, tumor apoptosis, and live cancer cells should help understand the establishment of tumor resistance of host immunity.

Reply: The reviewer raises an important question about the complexity of the tumor microenvironment, particularly how events in the tumors undergoing cell death compare to normal physiological cell death during homeostasis.  While it is clear that the mutational burden associated with tumors (both viable and apoptotic) will skew expression patterns of secreted contents and chemo-attractants, the main thesis developed in this paper is that molecular events associated with physiological/homeostatic apoptosis are hijacked or co-opted in the tumor microenvironment to foster an immune-compromised milieu.   Although how live tumor cells contribute to the microenvironment is outside the scope of this review, we have added a sentence on p9 that mutational burden will likely alter the scope of caspase mediated events, that caspase mediated events will be dynamic in the tumor microenvironment, as well as added a reference by Gregory and colleagues that also discuss cell death in normal versus neoplastic microenvironments.

Point #3: Anything known about differences in compensatory proliferative signals generated after radiation/chemo-drug treatments which induce massive death vs compensatory proliferative signals generated by spontaneous tumor cell death?

Reply:  While we discussed CRT, HMGB1, etc in the context of immunogenic death, this is an interesting but less researched area.  In the amended paper, we have added a statement that it is possible that unique AiPs are induced by radiation and chemotherapy, and referenced a recent study in an in vivo radiation model, whereby active PKC-z (zeta) induced by radiation correlates with increased proliferation (Wong et al, PLOS ONE, 2019). (p6, end of first paragraph).

Complexity of cell-cell interactions makes the figure 2 difficult to read.

Reply: Thank you for the suggestion.  We have reduced the numbers of cells in the figure to reduce the cluttering effect.

Check the spelling of all the text fields. There are some typos, for examples: “consititive” (line 541) and “cleanance” (line 565).

Reply: Thank you.  We have carefully reviewed the MS for typo’s and grammatical issues.

Reviewer 2 Report

Gadiyar et al. present a very detailed and well written review article on cell death in the microenvironment and possible strategies to exploit the pathways involved for cancer immunotherapy.

It would be nice to discuss the effects of apoptosis and efferocytosis on NK cells and the related suppression of anti-tumor immunity.

Author Response

Reviewer #2:

Reviewer #2 also had very supportive comments that “the paper presents a very detailed and well written review paper on cell death in the microenvironment and possible strategies to exploit the pathways involved in cancer immunotherapy?  Reviewer #2 also had a specific comment on what is known about efferocytosis on NK cells and the suppression of anti-tumor immunity.

Point #1” It would be nice to discuss the effects of apoptosis and efferocytosis on NK cells and the related suppression of anti-tumor immunity.

Reply:  This is an interesting point and while much is known about how efferocytosis effects macrophages and DCs, it appears much less is known on how NK cells are programmed natively or in the tumor microenvironment.  While more work will have to be done in this area, we have introduced this idea and cited a reference that efferocytosis induces macrophages to produce IL-4 and activate iNKT cells to resolve inflammation (Zeng et al, Blood, 2013).  This fits in well with recent results of Rothlin et al that apoptotic cells synergize with IL-4 to produce a repair program.  We also added the reference and a sentence in the Mertk section that PS receptors are important for NK mediated tumor killing and metastasis (Paolino et al, Nature 2014).

Reviewer 3 Report

This is a well structured, well written and illustrated review about how different modes of cell death lead to pro- or antitumorigenic effects, particularly modulation of the immune response. The authors were able to describe and discuss these complex interactions in a clear manner, and provided future perspectives for possible treatment options and difficulties that may be encountered. 

Minor comments:

Page 3,line 90: delete 'change'

page 7,line 277: 'undergoing during'; should be corrected

page 11, 417-422: very long sentence; grammar should be looked at.

page 11, line 424 (in title figure legend): 'promotes drive'; should be corrected

Author Response

Reviewer #3:

Reviewer #3 also had supportive comments “this is a well-structured and illustrated review about how different modes of cell death lead to pro- or anti-tumorogenic effects and had minor comments about typo’s and grammar.

Reply:  We thank the reviewer for his/her positive comments and have carefully edited the paper and corrected grammar mistakes.

Reviewer 4 Report

This is a good review and balanced assessment of the status of evidence that apoptosis and efferocytosis are exploited in cancer.

Apoptotic cell death is usually a response to the cell’s microenvironment and it is as fundamental to cellular and tissue physiology as cell division and differentiation. Attention to this form of cell death was prompted primarily by its crucial role in the normal embryonic development of higher vertebrates and in maintaining normal tissue homeostasis by controlling cell numbers and eliminating nonfunctioning, damaged, or misplaced cells. A now well-accepted concept is that apoptosis is an integral part of normal tissues functioning. There is no inflammatory response in apoptotic cells, and they are promptly ingested by professional cells such as macrophages or dendritic cells, but also by neighboring cells, i.e. non-professional cells, such as epithelial cells, endothelial cells, and fibroblasts (efferocytosis) in the tissue maintenance process. These phagocytes recognize and engulf apoptotic cells before their membrane is damaged, protecting surrounding tissues and cells from the damaging effects of the release of intracellular contents. If apoptotic cells are not ingested by phagocytes or epithelia, however, the cells proceed to a necrotic phase (called secondary necrosis), and their contents can spill into the extracellular space, causing inflammation and leading to inflammation-mediated tissue injury.

The authors, in this paper, summarize and examine deeper into, with knowledge and very clear description, not only the molecular and cellular mechanisms that trigger the removal and fate of apoptotic cells in normal condition, but also the involvement of alternative cell death forms through non-apoptotic processes in cancer progression. Moreover, they try, in the undeniably complex picture of the mechanisms involved in cell death, to understand the interplay between different cell death pathways, especially with a view toward targeting these pathways for cancer therapeutic purposes.

In fact, with this work, the authors' aim to elucidate the precise mechanisms behind cell death in the perspective of to develop new drugs anti-tumor. In many work focused on cell death, numerous cellular factors have been proposed to regulate cell death response following a variety of induction mechanisms in numerous cell types, but the role of many of these factors depends both on the signal triggering a given cell death process and on the type of cell in which the response is induced. In particular, in this review the authors focus attention on PS signalling as an anti-tumor therapeutic strategy and they summarize research on the roles of PS in physical and cancer biology.

I can conclude that the topic of the manuscript is inherent to the aim and scope of the journal, and that the design of the review is well done, and the paper clearly written.

Minor points to consider in subsequent versions.

The authors must check the references: i.e.

  • At pag 19, line 803 and 806: the reference: “Belzile, O., et al., Antibody targeting of phosphatidylserine for the detection and immunotherapy of cancer. Immunotargets Ther, 2018. 7: p. 1-14.” is reported twice: ref 123 and ref. 124. Remove one or add another one with the same statement.
  • At pag 19 line 816 and 821: the reference: “Gray, M.J., et al., Phosphatidylserine-targeting antibodies augment the anti-tumorigenic activity of anti-PD-1 therapy by enhancing immune activation and downregulating pro-oncogenic factors induced by T-cell checkpoint inhibition in murine triple-negative breast cancers. Breast Cancer Res, 2016. 18(1): p. 50.” is reported twice: ref 129 and ref. 131. Remove it or add another one with the same statement.

Author Response

Reviewer #4:

Reviewer #4 also had supportive comments “this is a good and well-balanced assessment of the status and evidence that apoptosis and efferocytosis are exploited in cancer” and also had relatively minor comments about typo’s and grammar, and duplicated references.

Reply:  We thank the reviewer for this attention to detail and have fixed reference duplications in the amended version.

We hope that we have addressed the suggested revisions to your satisfaction. Thank you again for your review of our manuscript that has improved both the clarity and content of the paper.  We hope that in its current revised state, it is acceptable for publication.